# Growth and trend analysis of area, production and yield of rice: A scenario of rice security in Bangladesh

Md. Abdullah Al Mamun[1]*, Sheikh Arafat Islam Nihad[2], Md. Abdur Rouf Sarkar[3], Md. Abdullah Aziz[1‡], Md. Abdul Qayum[1‡], Rokib Ahmed[1‡], Niaz Md Farhat Rahman[1‡], Md. Ismail Hossain[1‡], Md. Shahjahan Kabir[4‡]

1 Agricultural Statistics Division, Bangladesh Rice Research Institute, Gazipur, Bangladesh, 2 Plant Pathology Division, Bangladesh Rice Research Institute, Gazipur, Bangladesh, 3 Agricultural Economics Division, Bangladesh Rice Research Institute, Gazipur, Bangladesh, 4 Director General, Bangladesh Rice Research Institute, Gazipur, Bangladesh

☉ These authors contributed equally to this work.
‡ These authors also contributed equally to this work.
* mamunru4777@gmail.com

**Data Availability Statement:** All relevant data are within the paper and its Supporting Information files.

## Abstract

Bangladesh positioned as third rice producing country in the world. In Bangladesh, regional growth and trend in rice production determinants, disparities and similarities of rice production environments are highly desirable. In this study, the secondary time series data of area, production, and yield of rice from 1969–70 to 2019–20 were used to investigate the growth and trend by periodic, regional, seasonal and total basis. Quality checking, trend fitting, and classification analysis were performed by the Durbin-Watson test, Exponential growth model, Cochrane-Orcutt iteration method and clustering method. The production contribution to the national rice production of Boro rice is increasing at 0.97% per year, where Aus and Aman season production contribution significantly decreased by 0.48% and 0.49% per year. Among the regions, Mymensingh, Rangpur, Bogura, Jashore, Rajshahi, and Chattogram contributed the most i.e., 13.9%, 9.8%, 8.6%, 8.6%, 8.2%, and 8.0%, respectively. Nationally, the area of Aus and Aman had a decreasing trend with a -3.63% and -0.16% per year, respectively. But, in the recent period (Period III) increasing trend was observed in the most regions. The Boro cultivation area is increasing with a rate of 3.57% per year during 1984–85 to 2019–20. High yielding variety adoption rate has increased over the period and in recent years it has found 72% for Aus, 73.5% for Aman, and 98.4% for Boro season. As a result, the yield of the Aus, Aman, and Boro seasons has been found increasing growth for most of the regions. We have identified different cluster regions in different seasons, indicating high dissimilarities among the rice production regions in Bangladesh. The region-wise actionable plan should be taken to rapidly adopt new varieties, management technologies and extension activities in lower contributor regions to improve productivity. Cluster-wise, policy strategies should be implemented for top and less contributor regions to ensure rice security of Bangladesh.

**Funding:** The author(s) received no specific funding for this work.

**Competing interests:** The authors have declared that no competing interests exist.

## Introduction

Globally rice (*Oryza sativa*) is the third major cereal grain [1] and more than half of the world population consumes its as a staple food [2]. Broad adaptive capability in different ecosystems and less cultivation risk, several farmers preferred rice cultivation instead of other crops. World population is increasing and it is assumed that 14,886 million tons (MT) of foods need to be produced in 2050 to meet up the food demand [3]. Worldwide 503.17 MT rice is produced where China produces 29.5% of the total, followed by India (23.8%), Bangladesh (7.0%), Indonesia (6.9%), Vietnam (5.4%), and Thailand (3.7%) [4].

Rice is also the staple food in Bangladesh and accounting for approximately 78 percent of the country's total net cropped areas cultivation. The country achieves an autarky to meet up the rice demand for its 169.04 million peoples from 11.55 million hectares of cultivated gross area [5, 6]. In Bangladesh, food security is equivalent to rice security [5]. Rice is cultivated in three seasons namely Aus, Aman and Boro throughout the year. Since independence, rice production has been increased three-fold from approximately 11 MT in 1971–72 to about 36.6 MT in 2019–20 [7]. This revolution has transformed the country from so called "Bottomless Basket" to a "Full of Food Basket". After a long period, rice production in Bangladesh has risen significantly after 1990–1991, especially during two periods: 1996–1997 and 2000–01, as well as from 2009–10 to 2013–14. Improved loan distribution policies (credit deposits directly to farmers' 10 Taka bank accounts), well-organized fertilizer supplies, availability of high-quality seeds by the public and commercial sectors, and technical interventions (e.g. genetic improvements of varieties for favorable and unfavorable ecosystems) make it possible to make Bangladesh as one of the largest contributors of rice in the world [5, 8]. Bangladesh recently placed the third position worldwide in rice production, behind China and India, with a production volume of 3.6 crore tonnes [1].

In reality, the global food production has increased sharply since the Borlaug and Jennings days, keeping pace with an increasingly higher rate of population growth. To cite a country-specific example, we can easily refer to the Bangladesh scenario. Over the past four decades, Bangladesh succeeded in outpacing the population growth rate (1.3%) with its growth in rice output (2.8%) [7]. To increase or sustain the rice production, it is very important to interpose the rice-based technology in a specific ecosystems or specific locations. Knowledge about region specific rice cultivation scenario will be helpful to disseminate newly release technologies and to take necessary policy for sustainable rice production in Bangladesh. Therefore, this study investigates the growth and trend of area, production and yield of rice in Bangladesh based on periodic, regional, seasonal and total basis.

## Materials and methods

### Study area

Bangladesh is located in the northeastern part of South Asia. The majestic Himalayas stand some distance to the north, where the Bay of Bengal is in the south. These picturesque geographical boundaries frame a low-lying plain of about 1,47,570 square kilometers, criss-crossed by innumerable rivers and streams. The geographical extent of the country is between latitudes 20˚34' and 26˚38', and longitudes 88˚01' and 92˚41'. The country's borders are as follows: in the west, India (West Bengal); in the east, India (Tripura and Assam); in the south, Myanmar; and in the north, India (West Bengal and Meghalaya) [9]. Fig 1 depicts a conceptual framework of the study.

### Data used

Seasonal and total rice area, production and yield data were used from 1969–72 to 2019–20 at the national level (aggregate) and 1984–85 to 2019–20 at the regional level (disaggregate).

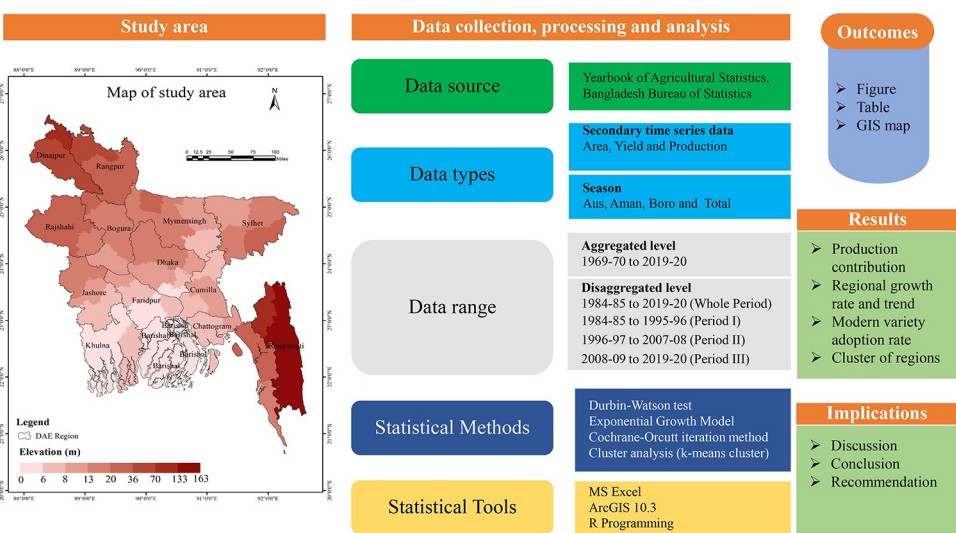

**Fig 1. A conceptual framework of the study.** Shapefile republished from Bangladesh Agricultural Research Council (BARC) database (http://maps.barcapps.gov.bd/index.php) under a CC BY license, with permission from Computer and GIS unit, BARC, original copyright 2014.

This study is designed based on secondary sources data published by the different issues of the Bangladesh Bureau of Statistics (BBS). For the analysis, region-wise time-series data were divided into three periods, Period I (1984–85 to 1995–96); Period II (1996–97 to 2007–08), and Period III (2008–09 to 2019–20). The regional variations were analyzed by considering an unchanging regional base of homogeneous environments. Region-wise scenarios would provide a base to explain the effects of specific conditions as well as agricultural development. In the published issues of the Year Book of Agricultural Statistics, district-wise rice data is available after 2006–07 [10]. Earlier area, production, and yield of rice data were published according to 23 crop production regions representing the 30 agro-ecological zones (AEZ) of Bangladesh. In this study, we aggregated all data into 14 agricultural regions because the Department of Agricultural Extension (DAE) conduct their activities according to these regions representing 64 districts of Bangladesh. The distribution of the studied regions is presented in Fig 1.

Geographic Information System (GIS) map was used to describe the regional variations of area, production, yield and total rice in Bangladesh. However, administrative shape file of Bangladesh was downloaded and used from website of the Bangladesh Agricultural Research Council. The specific link of the shape file ("Administrative map") is: http://maps.barcapps.gov.bd/index.php.

## Methods

The autocorrelation detection, regression model for growth estimations, and cluster analysis techniques were used in this study. In time series regression data, there are numerous causes of auto-correlation. Autocorrelation is often caused by the analyst's inability to incorporate one or more key predictor variables in the model. The existence of autocorrelation in the errors affects the ordinary least-squares regression method in many ways [11]. Although regression coefficients are still unbiased, they are no longer considered minimum-variance estimates. When errors are positively autocorrelated, the residual mean square may underestimate the error variance significantly. Autocorrelation may be detected using various statistical

**Table 1. Durbin-Watson statistic for detection of autocorrelation.**

| Region | Area | | | | Production | | | | Yield | | | |
|---|---|---|---|---|---|---|---|---|---|---|---|---|
| | Aus | Aman | Boro | Total | Aus | Aman | Boro | Total | Aus | Aman | Boro | Total |
| Barishal | 0.70 | 1.01 | 0.55 | 0.42 | 1.78 | 1.12 | 0.60 | 0.74 | 2.19 | 1.24 | 1.03 | 1.10 |
| Bogura | 0.18 | 1.18 | 0.20 | 0.51 | 0.53 | 1.09 | 0.62 | 0.99 | 2.02 | 0.69 | 1.68 | 1.07 |
| Chattogram | 0.50 | 1.48 | 0.77 | 0.93 | 1.23 | 1.08 | 1.41 | 1.04 | 1.80 | 1.56 | 1.46 | 1.89 |
| Cumilla | 0.59 | 1.17 | 0.27 | 1.23 | 1.27 | 1.22 | 0.36 | 0.64 | 2.05 | 0.91 | 0.71 | 0.66 |
| Dhaka | 0.22 | 1.43 | 0.24 | 1.13 | 0.78 | 1.41 | 0.30 | 0.66 | 1.63 | 1.29 | 0.61 | 0.64 |
| Dinajpur | 0.21 | 0.76 | 0.21 | 0.71 | 0.45 | 0.64 | 0.18 | 0.70 | 1.97 | 0.64 | 0.73 | 0.81 |
| Faridpur | 0.41 | 1.27 | 0.37 | 1.03 | 1.26 | 1.47 | 0.30 | 0.94 | 2.30 | 1.35 | 0.60 | 0.91 |
| Jashore | 0.31 | 1.34 | 0.28 | 1.81 | 1.27 | 1.31 | 0.31 | 0.84 | 2.27 | 1.37 | 0.70 | 0.99 |
| Khulna | 0.27 | 1.41 | 0.72 | 0.71 | 0.65 | 1.64 | 0.70 | 1.00 | 1.45 | 1.67 | 1.04 | 1.51 |
| Mymensingh | 1.37 | 1.12 | 0.56 | 0.80 | 2.06 | 1.31 | 0.63 | 0.78 | 1.94 | 1.40 | 0.97 | 0.91 |
| Rajshahi | 0.23 | 1.35 | 0.18 | 0.81 | 1.30 | 0.86 | 0.16 | 0.52 | 2.30 | 0.60 | 0.96 | 0.33 |
| Rangamati | 0.60 | 0.46 | 0.54 | 0.39 | 0.80 | 0.50 | 0.41 | 0.41 | 1.67 | 1.15 | 0.56 | 0.84 |
| Rangpur | 0.21 | 1.15 | 0.34 | 0.37 | 0.24 | 1.48 | 0.63 | 1.20 | 1.73 | 1.77 | 1.44 | 1.24 |
| Sylhet | 0.46 | 1.48 | 2.11 | 1.17 | 1.03 | 1.91 | 1.77 | 1.64 | 2.44 | 1.15 | 0.89 | 0.72 |
| Bangladesh | 0.21 | 1.56 | 0.22 | 0.74 | 1.29 | 1.43 | 0.26 | 0.76 | 2.33 | 1.18 | 0.58 | 0.70 |

techniques. Durbin and Watson [12] developed a procedure that is frequently utilized world-wide. This test assumes that the errors in the regression model are produced by a first-order autoregressive process observed at evenly spaced time intervals [11]. A significant Durbin-Watson statistic or a suspicious residual plot suggests that autocorrelated model errors may be present. Table 1 represent the value of Durbin-Watson statistics and found that the data have positive autocorrelation. It may be due to a real-time dependency in error or an 'artificial' time dependence induced by the absence of one or more key predictor variables.

The observed autocorrelation in the model errors cannot be eliminated by adding additional predictor variables to the model. The autocorrelative structure in the model must be explicitly considered, and an appropriate parameter estimation technique must be used. The method developed by Cochrane and Orcutt [13] is a very excellent and frequently utilized approach. We used the linear model suggested by Finger [14] for analysis of the production contribution trend and expressed as:

$$Y_t = \beta_0 + \beta_1 t + e_t \tag{1}$$

where,
$Y_t$ = Expected value at t
t = Time index
$\beta_0$ = Model intercept
$\beta_1$ = Regression coefficient
e = Residual
Also, exponential models were employed to assess the growth analysis. The exponential growth trend model and the equation of the model is as follows:

$$Y_t = ae^{bt} \tag{2}$$

$Y_t$ = Area, production, and yield of rice in year t
a = Model intercept
b = Annual rate of change of rice area, production, and yield
e = Residual

The cluster analysis is then used to find geographical groupings of rice-producing areas with comparable features in terms of production growth and adoption of high-yielding varieties. The variables used in the cluster analysis is independent. Thus, the cluster analysis uses the distinct principal components proposed by Huth and Pokorná [15]. Applying the most commonly used non-hierarchical clustering technique, K-means clustering [16, 17] that classify the 14 agricultural regions into K clusters using Euclidean distance as the linkage method. For all of the analyses, we used Microsoft Excel, ArcGIS 10.3, and R programming tools.

## Results

### Historical (1969–70 to 2019–20) rice area coverage and production contribution by seasons

The area coverage and production contribution of Aus, Aman and Boro rice have been shown in Fig 2. We found that historically the area coverage of Boro rice has increased significantly whereas the area of Aus and Aman season significantly decreased from 1969–70 to 2019–20. On the other hand, the production contribution of the Boro rice has increased from 16.1% to 53.7% and the increasing rate is 0.97% per year. Aus season production contribution has

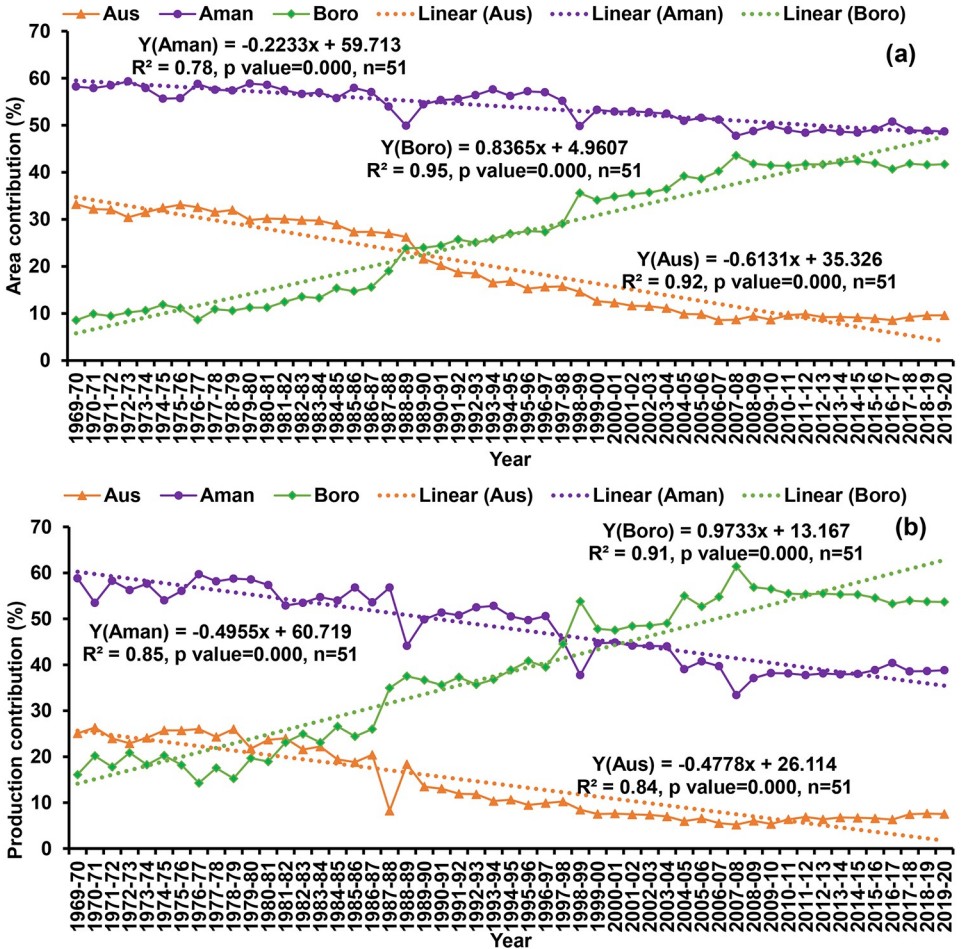

**Fig 2.** National area coverage (a) and production contribution (b) of Aus, Aman and Boro rice during 1969–70 to 2019–20.

significantly decreased from 25.1% in 1969–70 to 7.5% in 2019–20 and the decreased rate was 0.48% per year. Production contribution of the Aman season has been significantly decreased 58.8% to 38.8% with a decreasing rate of 0.49% per year. Among the rice season, Aus production contribution is drastically reduced compared to the Aman season.

## Region-wise area and production of rice cultivation in Bangladesh

Thirty-six years (1984–85 to 2019–20) region-wise area and production of rice cultivation in Bangladesh is depicted in Fig 3. Periodic production contributions follow an increasing trend in Dinajpur, Mymensingh, Rajshahi and Rangamati where a periodic decreasing trend was observed in Chattogram and Cumilla regions (Fig 3A). In the recent period, the production contribution of Dhaka, Bogura, Jashore and Faridpur is decreased compared to the earlier

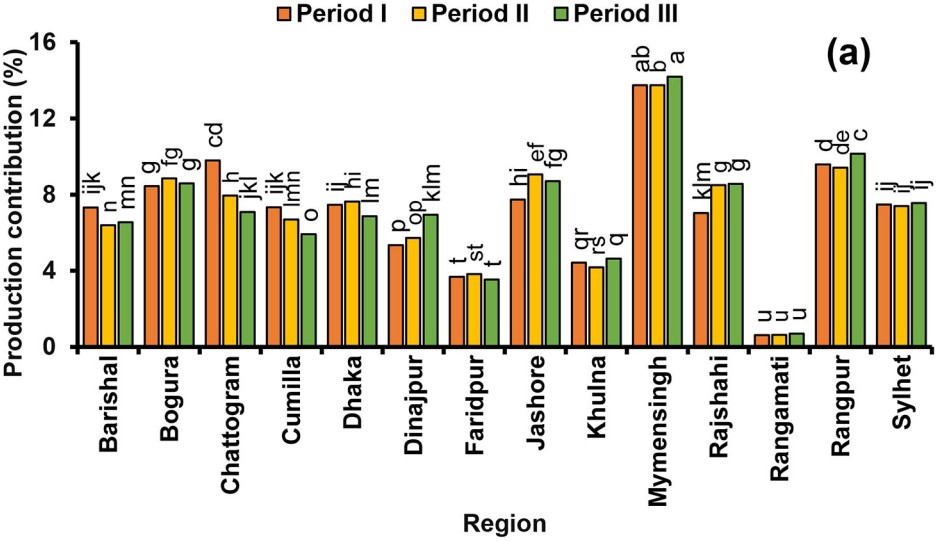

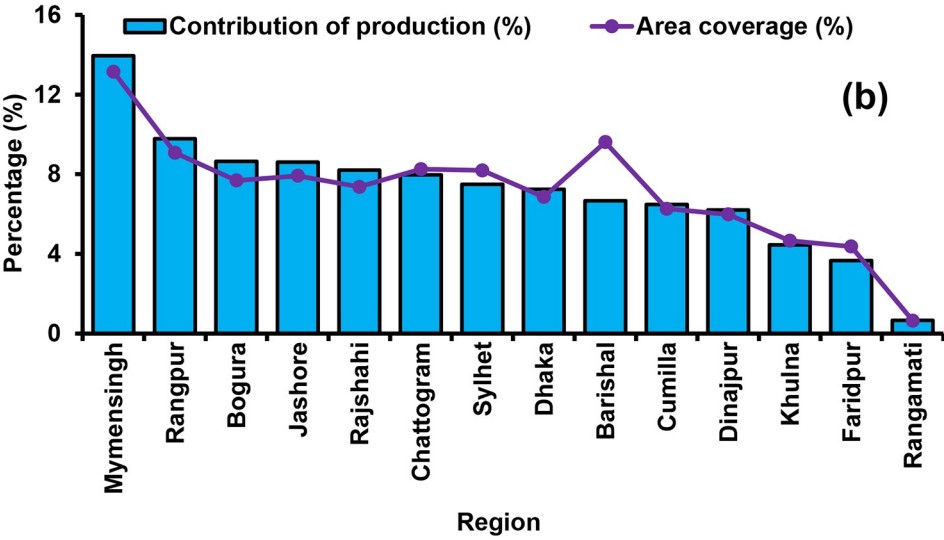

**Fig 3.** Periodic production contribution of major rice-growing regions (a), and regional production contribution and area coverage of rice (b) during 1984–85 to 2019–20. Period I: 1984–85 to 1995–96, Period II: 1996–97 to 2007–08, Period III: 2008–09 to 2019–20.

period. Based on historical production trend, Mymensingh, Rangpur, Bogura, Jashore, Rajshahi and Chattogram significantly contributed 13.9%, 9.8%, 8.6%, 8.6%, 8.2% and 8.0%, respectively of the total production, and they were positioned as the most rice contributed regions compared to others. On the other hand, Khulna (4.4%) and Faridpur (3.7%) contributed less compared to other regions and Rangamati was the lowest production contributor (0.7%) among all regions. However, the rice cultivation area of Mymensingh has been found as higher compared to other regions where Rangamati had the lowest rice cultivation area in Bangladesh (Fig 3B).

## Regional trend and growth analysis of rice

Spatial and temporal (periodic) variability has been found in the rice cultivation area of Bangladesh (Fig 4). Over the period, regional variation prevails in terms of rice cultivation area (Fig 5 and Table 2). A continuous decreasing trend was observed in Aus cultivated area and an increasing trend was found in all regions for Boro season. The highest area decreasing rate (-18.43%) was found in Rangpur throughout the period in Aus season but in the recent period, the area is increasing at the rate of 29.62%. However, the Aman rice area fluctuated slightly in all regions, but Dhaka, Faridpur, Khulna and Sylhet regions showed a significantly decreasing trend (-0.43 to -1.54%). Overall Bangladesh, the Aus and Aman area decreased by -3.63% and -0.16% per year, respectively and the Boro area is increasing with an annual rate of 3.57% during 1984–85 to 2019–20. In Period I and Period II, cultivated areas were decreased in most of the regions for the Aus and Aman season, but in the recent period (Period III), i.e., an increasing trend was observed in most of the regions. In total, the highest growth rate of rice area was found in Rangamati (2.03%) and the lowest was observed in Faridpur (-1.25%).

Spatial and temporal (periodic) variability of rice production were observed in Bangladesh (Fig 6). Rice production trends of Aus, Aman, Boro and total are presented in Fig 7 and Table 3. The long-term production trend was found significantly increasing (2.18% to 10.25%) in all regions for the Boro season. In Aman season, production trends of all regions except Cumilla (-0.06%) were significantly increased (0.95% to 3.61%) over the period. Significant decreasing production trend (-15.33% to -4.05%) was found in most of the region for the Aus

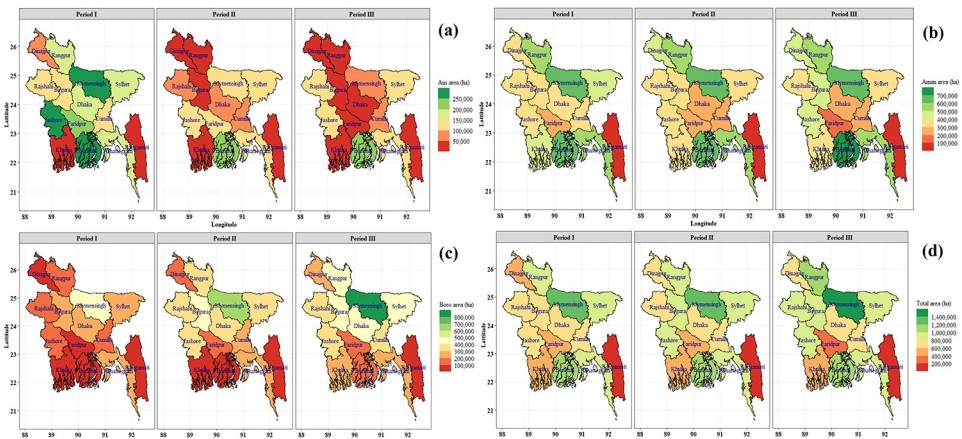

**Fig 4.** Spatial (regional) and temporal (periodic) distribution of Aus (a), Aman (b), Boro (c) and total (d) rice cultivation area of Bangladesh. Period I: 1984–85 to 1995–96, Period II: 1996–97 to 2007–08, Period III: 2008–09 to 2019–20. Shapefile republished from Bangladesh Agricultural Research Council (BARC) database (http://maps. barcapps.gov.bd/index.php) under a CC BY license, with permission from Computer and GIS unit, BARC, original copyright 2014.

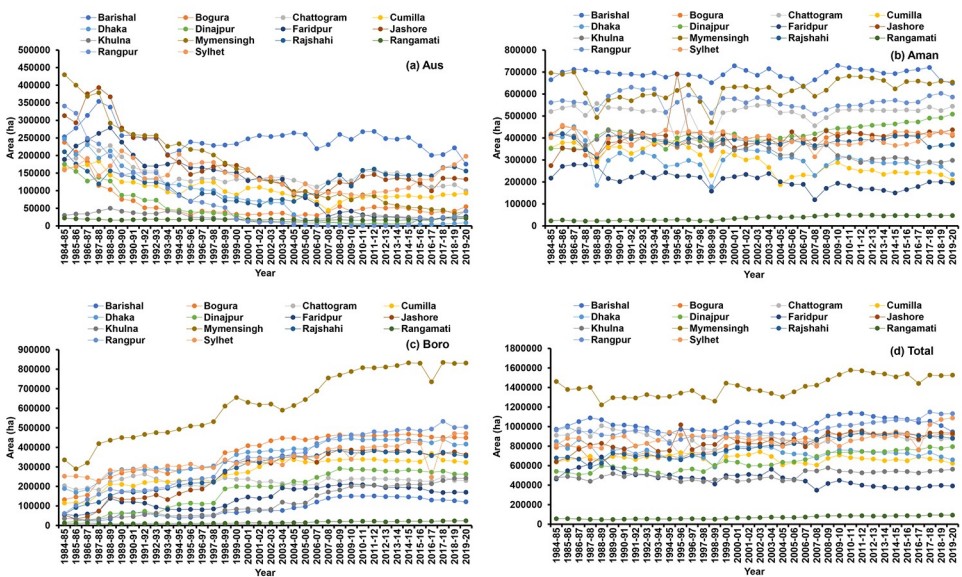

**Fig 5.** Trend of Aus (a), Aman (b), Boro (c) and total (d) rice cultivation area in Bangladesh.

season but Barishal (2.39%), Rajshahi (3.49%) and Sylhet (1.07%) showed a significant increasing trend. In Aus, Aman and Boro season, the highest production growth rate was found in Rajshahi (3.49%), Rangamati (3.61%) and Dinajpur (10.25%), respectively. In total, the highest production growth rate was found in Dinajpur (4.07%) and the lowest was observed in Chattogram (1.64%).

**Table 2. Periodic growth rate (%) of Aus, Aman, Boro and total rice cultivation area at the regional level.**

| Region | Aus | | | | | Aman | | | | | Boro | | | | | Total | | | | |
|---|---|---|---|---|---|---|---|---|---|---|---|---|---|---|---|---|---|---|---|---|
| | PI | PII | PIII | WP | CV (%) | PI | PII | PIII | WP | CV (%) | PI | PII | PIII | WP | CV (%) | PI | PII | PIII | WP | CV (%) |
| Barishal | -0.96 | 1.63** | -1.15 | -0.43* | 14.03 | 0.41 | 0.12 | 0.60 | 0.10 | 3.25 | 3.05 | 8.16** | -0.10 | 4.77** | 49.58 | -0.75 | 0.58 | -1.31* | 0.20* | 5.56 |
| Bogura | -18.29** | -2.03* | 0.55 | -4.17** | 80.86 | -2.24 | 0.68 | 0.71 | 0.14 | 8.33 | 10.56** | 4.80** | 0.01 | 3.76** | 28.73 | -1.07 | 1.76* | -0.13 | 0.96* | 11.63 |
| Chattogram | -3.90* | -0.11 | -0.44 | -1.41** | 23.11 | 0.28 | 1.99 | 0.91* | 0.04 | 8.06 | 3.27** | 0.35 | 0.38 | 0.52* | 8.58 | -1.57* | 0.24 | -0.19 | -0.12 | 6.78 |
| Cumilla | -6.27** | -1.96 | 3.58* | -2.58** | 30.26 | -0.57 | -4.93** | -0.29 | -1.54** | 20.43 | 7.82** | 4.21** | 0.31 | 3.15** | 26.59 | 0.86* | -0.92 | -0.59* | -0.10 | 5.38 |
| Dhaka | -7.95** | -13.12** | -8.52** | -13.84** | 97.65 | -3.91 | 2.29 | 0.29 | -0.80* | 18.11 | 5.04** | 3.46** | 0.13 | 2.61** | 23.84 | -1.81* | 0.50 | -0.86* | -0.21 | 7.50 |
| Dinajpur | -10.86** | -16.89** | 8.20 | -10.18** | 110.96 | 1.91** | 0.45 | 1.39** | 0.75** | 10.17 | 19.64** | 7.45** | 0.40 | 8.58** | 53.78 | 0.03 | 1.50* | 0.72* | 1.29* | 14.68 |
| Faridpur | -2.16 | -6.20** | -8.62** | -8.23** | 70.13 | -2.22* | -0.82 | 0.84 | -1.48** | 18.03 | 9.36** | 9.29** | -1.44** | 4.12** | 36.48 | -1.35 | -1.04 | -0.99* | -1.25* | 16.52 |
| Jashore | -4.21** | -5.21** | 2.01 | -3.64** | 47.13 | 2.96** | -2.24 | 0.39 | 0.60** | 15.15 | 16.43** | 7.33** | 0.31 | 6.68** | 44.71 | 2.02* | 0.35 | -0.18 | 0.71* | 9.69 |
| Khulna | -1.01 | -9.30** | -0.07 | -2.23** | 37.50 | 0.09 | -1.23** | -2.08** | -1.03** | 12.44 | 8.94** | 8.62** | 2.67** | 6.94** | 63.24 | -0.02 | 1.38* | -0.03 | 0.49* | 8.40 |
| Mymensingh | -6.74** | -8.49** | -6.70** | -6.71** | 67.70 | -2.21* | 0.52 | 0.40 | 0.11 | 8.13 | 5.26** | 2.24** | 0.95* | 2.97** | 27.09 | -0.69 | 0.35 | -0.14 | 0.48* | 7.08 |
| Rajshahi | -4.35** | -1.20 | 2.97* | -0.37 | 31.76 | -0.71 | -0.12 | 0.43 | -0.09 | 5.32 | 12.71** | 5.78** | -0.07 | 5.01** | 36.52 | 0.39 | 1.73* | -0.09 | 1.17* | 13.03 |
| Rangamati | -3.11** | -1.03 | 2.13* | -0.56** | 11.05 | -1.66 | 5.08** | 0.70 | 2.40** | 29.39 | -1.82 | 3.83** | 1.23* | 2.84** | 35.69 | -0.01 | 3.56* | 0.74* | 2.03* | 22.74 |
| Rangpur | -11.28** | -32.49** | 29.62* | -18.43** | 132.46 | 1.19** | 0.11 | 0.94** | -0.10 | 5.28 | 13.45** | 4.94** | 1.58** | 5.71** | 41.75 | -0.93* | 0.51 | 1.29* | 0.52* | 7.87 |
| Sylhet | -2.45 | -5.72** | 4.48** | -1.87** | 26.28 | -1.41 | -1.40** | 0.78 | -0.43** | 7.98 | 1.99* | 1.26** | 0.00 | 1.46** | 19.51 | 0.72 | -1.12* | 1.99* | 0.28* | 8.19 |
| Bangladesh | -5.91** | -4.36** | 0.64 | -3.63** | 41.75 | -0.43 | -0.20 | 0.48 | -0.16* | 4.23 | 6.87** | 4.28** | 0.50 | 3.57** | 30.57 | -0.36* | 0.48 | 0.03 | 0.39* | 5.18 |

Note. Period I (PI): 1984–85 to 1995–96, Period II (PII): 1996–97 to 2007–08, Period III (PIII): 2008–09 to 2019–20, Whole Period (WP): 1984–85 to 2019–20, CV: Coefficient of variation (whole period)

**Significant at 1% level

*Significant at 5% level.

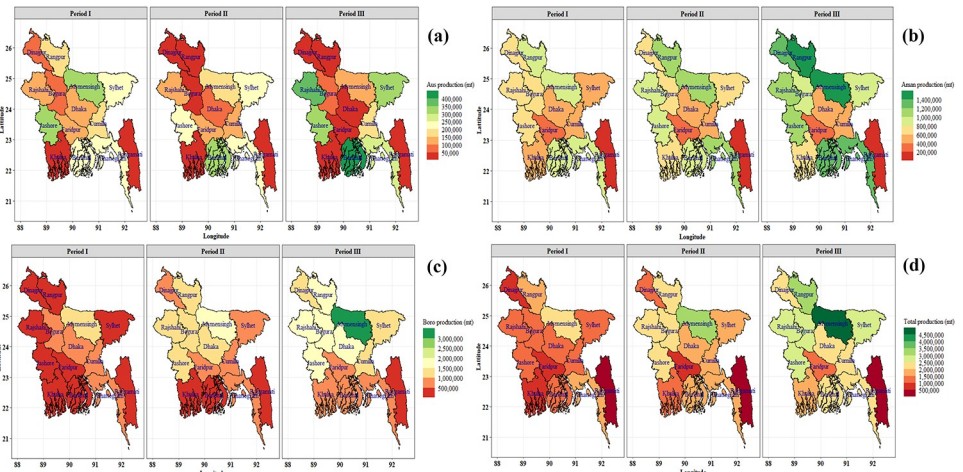

**Fig 6.** Spatial (regional) and temporal (periodic) distribution of Aus (a), Aman (b), Boro (c) and total (d) rice production (metric ton) of Bangladesh. Period I: 1984–85 to 1995–96, Period II: 1996–97 to 2007–08, Period III: 2008–09 to 2019–20. Shapefile republished from Bangladesh Agricultural Research Council (BARC) database (http://maps.barcapps.gov.bd/index.php) under a CC BY license, with permission from Computer and GIS unit, BARC, original copyright 2014.

Spatial and temporal (periodic) variability of rice yield were observed in Bangladesh (Fig 8). The yield trend was found increasing for all the regions and seasons of Bangladesh but region to region yield variations was very high (Fig 9 and Table 4). In most of the region, a significant increasing growth rate was found for Aus, Aman, Boro and total rice yield. In Aus, the highest annual growth rate of yield was found in Rajshahi (3.86%) and the lowest was in Rangamati (0.89%). In Aman, the highest growth rate of yield was observed in Faridpur (2.95%) and the lowest was in Rangamati (1.21%). In Boro, the highest growth rate of yield was found in Sylhet (2.78%) and the lowest was observed in Bogura (1.30%). In total, the highest growth rate of yield was found in Faridpur (4.09%) and the lowest was observed in Rangamati (1.38%).

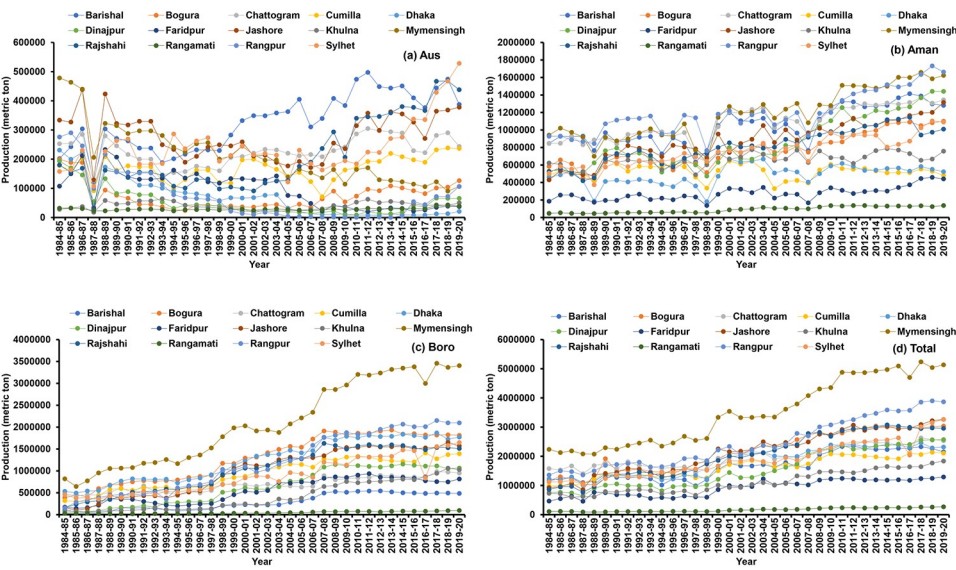

**Fig 7.** Trend of Aus (a), Aman (b), Boro (c) and total (d) rice production in Bangladesh.

**Table 3. Periodic growth rate (%) of Aus, Aman, Boro and total rice production at the regional level.**

| Region | Aus | | | | | Aman | | | | | Boro | | | | | Total | | | | |
|---|---|---|---|---|---|---|---|---|---|---|---|---|---|---|---|---|---|---|---|---|
| | PI | PII | PIII | WP | CV (%) | PI | PII | PIII | WP | CV (%) | PI | PII | PIII | WP | CV (%) | PI | PII | PIII | WP | CV (%) |
| Barishal | -0.58 | 6.58** | 1.89 | 2.39** | 29.64 | 1.08 | 3.81* | 3.78** | 1.54** | 20.57 | 4.73 | 11.06** | 0.78 | 6.66** | 63.24 | -0.14 | 3.77* | 0.27 | 2.42* | 27.90 |
| Bogura | -17.91** | 2.74 | 6.44* | -0.93 | 62.13 | 1.63 | 3.13** | 3.33** | 1.89** | 24.21 | 10.68** | 6.73** | 0.07 | 5.06** | 40.16 | 2.39 | 4.99* | 0.80* | 3.09* | 31.85 |
| Chattogram | -3.89 | 3.87** | 1.56 | 0.59 | 20.85 | 2.05** | 3.04 | 2.40** | 1.53** | 17.99 | 2.15* | 2.82* | 1.85* | 2.18** | 25.02 | 0.55 | 2.38* | 1.19* | 1.64* | 18.81 |
| Cumilla | -3.15 | 2.88 | 7.26** | 0.14 | 24.30 | 1.80 | -2.17 | 1.39 | -0.06 | 15.30 | 8.38** | 7.03** | 0.63 | 4.48** | 38.57 | 3.19* | 2.65* | 0.44 | 2.15* | 22.87 |
| Dhaka | -9.43* | -8.61* | -1.68 | -10.91** | 95.52 | -2.44 | 6.03 | 1.32 | 1.10* | 23.30 | 4.78** | 6.80** | 1.14* | 4.17** | 38.62 | 0.90 | 4.68* | 0.07 | 2.59* | 27.91 |
| Dinajpur | -10.79** | -7.88** | 11.31* | -6.70** | 88.81 | 4.66** | 3.04* | 3.41** | 2.76** | 34.53 | 20.51** | 10.27** | 0.87 | 10.25** | 65.44 | 2.70* | 6.12* | 1.38* | 4.07* | 42.93 |
| Faridpur | 0.13 | -4.83* | -2.28 | -5.89** | 65.11 | -0.18 | 3.16 | 4.79** | 1.46** | 28.12 | 9.31** | 13.35** | -0.32 | 5.93** | 50.35 | 1.06 | 5.51* | 0.30 | 2.84* | 30.67 |
| Jashore | 0.04 | -1.58 | 5.57** | -0.10 | 27.67 | 7.48** | 2.53* | 2.15* | 2.78** | 25.75 | 16.99** | 10.08** | 1.09* | 8.03** | 53.48 | 4.93* | 3.72* | 1.27* | 3.56* | 34.63 |
| Khulna | 3.51 | -7.81** | 3.78 | -0.06 | 33.34 | 2.15* | 2.22* | 0.28 | 0.95** | 12.68 | 9.79** | 11.15** | 3.80** | 9.11** | 80.35 | 1.84 | 4.75* | 2.04* | 3.19* | 34.92 |
| Mymensingh | -6.00** | -3.93** | -2.81 | -4.05** | 46.39 | 0.18 | 3.64* | 2.95** | 1.92** | 24.21 | 6.46** | 5.21** | 2.32** | 5.00** | 45.38 | 1.47 | 3.61* | 1.30* | 3.04* | 32.25 |
| Rajshahi | 0.02 | 2.24 | 6.59** | 3.49** | 57.19 | 3.85* | 3.24** | 1.85* | 2.34** | 23.90 | 14.12** | 7.78** | 0.20 | 6.71** | 48.76 | 5.10* | 4.97* | 0.64* | 3.89* | 37.87 |
| Rangamati | -0.64 | 0.70 | 2.96** | 0.33 | 16.02 | 0.65 | 7.03** | 1.96* | 3.61** | 38.68 | -3.02* | 6.44** | 2.30** | 4.30** | 53.89 | 0.41 | 5.66* | 1.42* | 3.41* | 36.92 |
| Rangpur | -8.53** | -28.92** | 37.56** | -15.33** | 113.89 | 2.98** | 1.40 | 3.77** | 1.50** | 20.59 | 12.35** | 6.59** | 2.13** | 7.53** | 57.07 | 2.04* | 3.16* | 2.42* | 3.17* | 33.96 |
| Sylhet | 1.46 | -3.64* | 8.06** | 1.07* | 36.71 | 1.46 | 1.61* | 1.42 | 1.84** | 22.48 | 2.07 | 5.70** | 2.19 | 4.24** | 46.41 | 2.78* | 2.21* | 2.75* | 3.00* | 32.78 |
| Bangladesh | -3.98 | -0.37 | 4.40** | -0.60 | 22.55 | 2.14** | 2.69* | 2.59** | 1.73** | 20.22 | 7.55** | 7.17** | 1.36* | 5.41** | 46.54 | 2.13* | 3.92* | 1.23* | 2.96* | 30.88 |

Note. Period I (PI): 1984–85 to 1995–96, Period II (PII): 1996–97 to 2007–08, Period III (PIII): 2008–09 to 2019–20, Whole Period (WP): 1984–85 to 2019–20, CV: Coefficient of variation (whole period).

**Significant at 1% level

*Significant at 5% level.

## Modern variety adoption

Periodic modern varieties adoption (%) in different regions are illustrated in Fig 10. In Aus season, high yielding varieties (HYVs) adoption was very low in Periods I and II, but in recent years adoption is much higher than earlier and it is gradually increasing. For the Aman season, the HYVs adoption percentage is gradually increased in all regions, while the lowest adoption was found in Barishal and Faridpur regions. HYVs adoption in Boro season has found always been higher than Aus and Aman. Though adoption is higher in Boro season from the earlier periods, it is still gradually increasing and it reaches almost 100% in most of the areas. However, the adoption rate was comparatively low in Sylhet region during Boro season. Over the

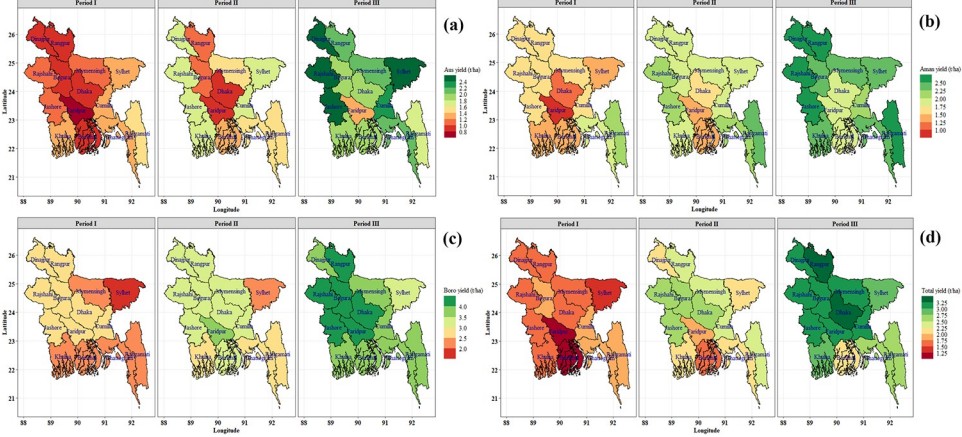

**Fig 8.** Spatial (regional) and temporal (periodic) distribution of Aus (a), Aman (b), Boro (c) and total (d) rice yield (metric ton) of Bangladesh. Period I: 1984–85 to 1995–96, Period II: 1996–97 to 2007–08, Period III: 2008–09 to 2019–20. Shapefile republished from Bangladesh Agricultural Research Council (BARC) database (http://maps.barcapps.gov.bd/index.php) under a CC BY license, with permission from Computer and GIS unit, BARC, original copyright 2014.

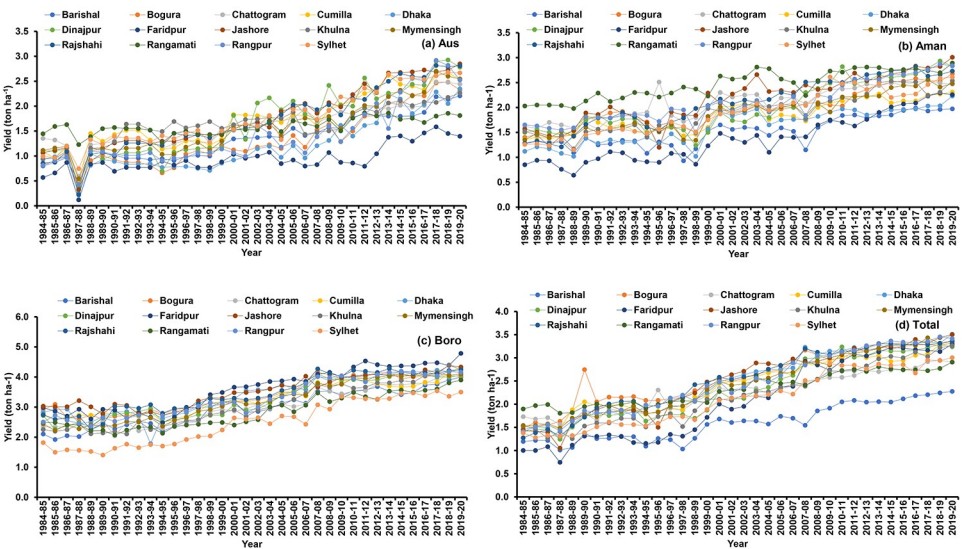

**Fig 9.** Trend of Aus (a), Aman (b), Boro (c) and total (d) rice yield in Bangladesh.

period, the modern variety adoption percentage gradually increasing in all over Bangladesh. The countrywide average modern variety adoption (%) in different seasons are shown in Fig 11. In recent period, the adoption percentage of Aus rice reached 72% from 21.4%. In Period I, the HYV adoption percentage for Aman season was 34.15% and it increased by 73.5% in Period III. Similarly, for Boro season adoption is gradually increasing and it reaches 88.6% to 98.4% from Period I to Period III.

## Cluster analysis for identifying the dynamics of rice varietal adoption and production growth

The cluster analysis is used to identify groups of identical rice production regions depicting similar characteristics in their modern variety adoption and production growth rate (Fig 12).

**Table 4. Periodic growth rate (%) of Aus, Aman, Boro and total rice yield at the regional level.**

| Region | Aus | | | | | Aman | | | | | Boro | | | | | Total | | | | |
|---|---|---|---|---|---|---|---|---|---|---|---|---|---|---|---|---|---|---|---|---|
| | PI | PII | PIII | WP | CV (%) | PI | PII | PIII | WP | CV (%) | PI | PII | PIII | WP | CV (%) | PI | PII | PIII | WP | CV (%) |
| Barishal | 0.38 | 4.96** | 3.05** | 2.83** | 33.35 | 0.67 | 3.66** | 3.22** | 1.43** | 19.54 | 1.68* | 2.90** | 0.89* | 1.89** | 20.67 | 0.62 | 3.19* | 1.58* | 2.22* | 24.84 |
| Bogura | 0.37 | 4.77** | 5.90** | 3.25** | 41.44 | 3.86** | 2.44** | 2.61** | 1.75** | 19.44 | 0.12 | 1.93** | 0.06 | 1.30** | 16.91 | 3.47* | 3.24* | 0.93* | 2.13* | 22.41 |
| Chattogram | 0.00 | 3.98** | 2.00** | 2.00** | 25.66 | 1.77** | 1.05 | 1.49** | 1.50** | 15.84 | -1.12 | 2.46** | 1.47** | 1.66** | 20.83 | 2.12* | 2.14* | 1.38* | 1.76* | 18.66 |
| Cumilla | 3.11 | 4.84** | 3.68** | 2.72** | 31.46 | 2.35** | 2.79** | 1.68** | 1.49** | 16.97 | 0.56 | 2.83** | 0.32 | 1.34** | 15.99 | 2.33* | 3.57* | 1.03* | 2.25* | 23.58 |
| Dhaka | -1.48 | 4.51** | 6.83** | 2.93** | 43.71 | 1.45 | 3.75* | 1.06** | 1.90** | 21.66 | -0.26 | 3.35** | 1.01** | 1.56** | 17.89 | 2.72* | 4.18* | 0.94* | 2.81* | 28.55 |
| Dinajpur | 0.07 | 9.00** | 3.11** | 3.48** | 40.38 | 2.76** | 2.59* | 2.02** | 2.01** | 24.99 | 0.88 | 2.82** | 0.46 | 1.66** | 19.81 | 2.67* | 4.63* | 0.65* | 2.77* | 29.47 |
| Faridpur | 2.29 | 1.37 | 6.34** | 2.34** | 30.79 | 2.08 | 3.94** | 3.97** | 2.95** | 34.56 | -0.06 | 4.06** | 1.12** | 1.81** | 19.91 | 2.42 | 6.55* | 1.30* | 4.09* | 41.57 |
| Jashore | 4.26 | 3.63** | 3.56** | 3.54** | 35.34 | 4.53** | 4.77** | 1.76** | 2.19** | 21.79 | 0.56 | 2.75** | 0.78** | 1.35** | 14.79 | 2.91 | 3.37* | 1.45* | 2.85* | 28.58 |
| Khulna | 4.52 | 1.49** | 3.85** | 2.17** | 24.27 | 2.07* | 3.44** | 2.36** | 1.97** | 20.95 | 0.85 | 2.53** | 1.13** | 2.17** | 23.76 | 1.86* | 3.37* | 2.07* | 2.70* | 28.46 |
| Mymensingh | 0.74 | 4.56** | 3.90** | 2.65** | 32.95 | 2.41** | 3.11** | 2.53** | 1.81** | 19.81 | 1.20** | 2.97** | 1.37* | 2.03** | 21.20 | 2.16* | 3.26* | 1.44* | 2.56* | 26.43 |
| Rajshahi | 4.38 | 3.44** | 3.62** | 3.86** | 37.71 | 4.57** | 3.35** | 1.43** | 2.44** | 23.43 | 1.41 | 2.00** | 0.27 | 1.70** | 18.27 | 4.71* | 3.24* | 0.73* | 2.72* | 26.98 |
| Rangamati | 2.47* | 1.73** | 0.83 | 0.89** | 10.20 | 2.31** | 1.94** | 1.27* | 1.21** | 11.68 | -1.20* | 2.61** | 1.07** | 1.47** | 19.57 | 0.42 | 2.09* | 0.67* | 1.38* | 15.15 |
| Rangpur | 2.75 | 3.58** | 7.94** | 3.10** | 43.78 | 1.80** | 1.28 | 2.83** | 1.61** | 19.47 | -1.10 | 1.65** | 0.56* | 1.82** | 20.84 | 2.97* | 2.65* | 1.12* | 2.65* | 27.15 |
| Sylhet | 3.91** | 2.07* | 3.58** | 2.95** | 31.79 | 2.86** | 3.01** | 0.64 | 2.27** | 23.54 | 0.07 | 4.45** | 2.19** | 2.78** | 29.87 | 2.06* | 3.32* | 0.76* | 2.72* | 28.36 |
| Bangladesh | 1.93 | 3.99** | 3.76** | 3.03** | 34.34 | 2.57** | 2.90** | 2.11** | 1.89** | 20.16 | 0.69* | 2.89** | 0.86** | 1.84** | 19.73 | 2.49* | 3.44* | 1.20* | 2.57* | 26.40 |

Note. Period I (PI): 1984–85 to 1995–96, Period II (PII): 1996–97 to 2007–08, Period III (PIII): 2008–09 to 2019–20, Whole Period (WP): 1984–85 to 2019–20, CV: Coefficient of variation (whole period).

**Significant at 1% level

*Significant at 5% level.

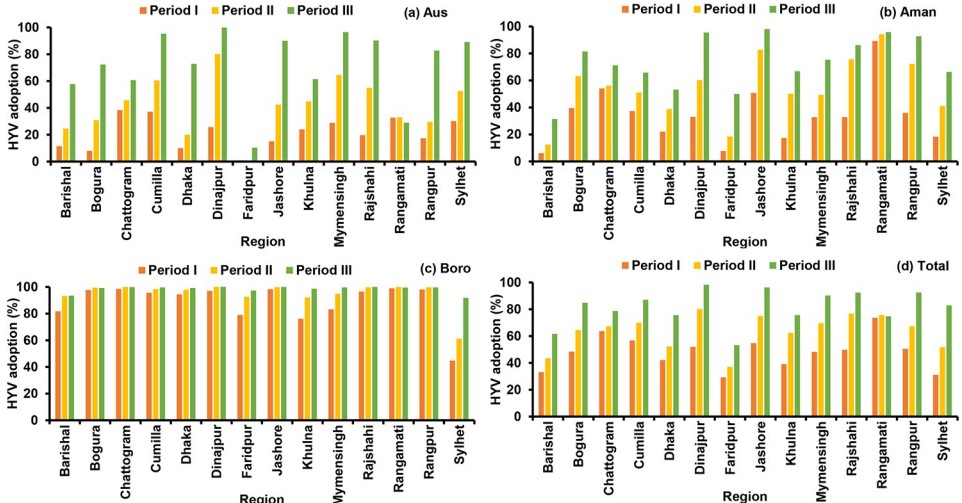

**Fig 10.** Periodic and regional high yielding varieties adoption (%) of Aus (a), Aman (b), Boro (c) and total (d) rice season in Bangladesh. Period I: 1984–85 to 1995–96, Period II: 1996–97 to 2007–08, Period III: 2008–09 to 2019–20.

From the analysis, different classes of cluster have been identified to classify the regions exhibiting significant rising trends, significant decreasing trends and mixed or insignificant trends in the rice production and high yielding adoption growth rates. Fig 12 is prepared by using K-means clustering to reflect the aforesaid distinct characteristics. In Aus season, HYV variety adoption growth rate is positive for all the regions except Rangamati. But in Mymensingh, Bogura, Dinajpur, Rangpur, Dhaka and Faridpur have negative production growth rate and formed the same cluster. For Aman season, we have identified four clusters where Dinajpur, Rajshahi, Sylhet, Barishal formed a similar cluster. Another cluster contains Jashore,

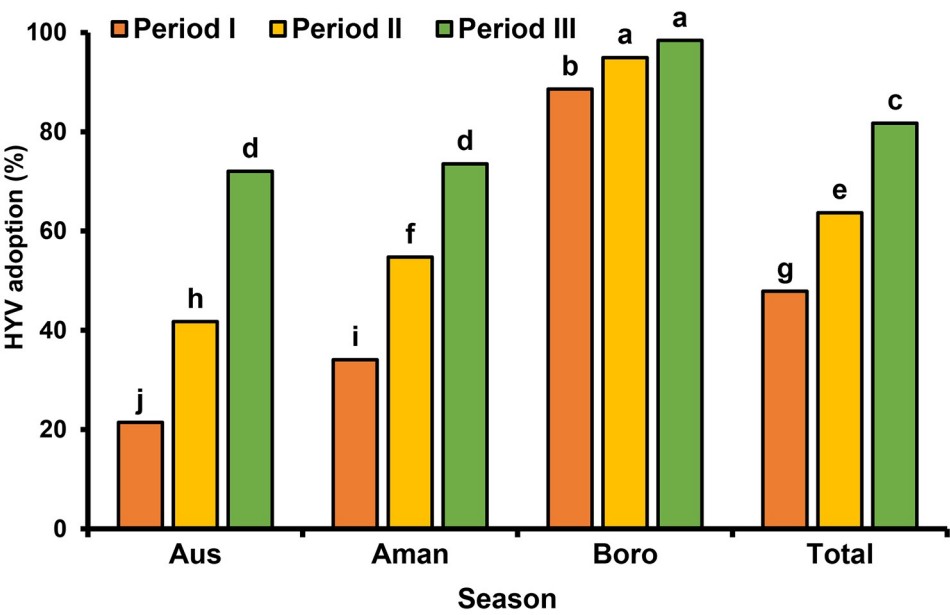

**Fig 11. Seasonal and total high yielding varieties adoption (%) in Bangladesh.** Period I: 1984–85 to 1995–96, Period II: 1996–97 to 2007–08, Period III: 2008–09 to 2019–20.

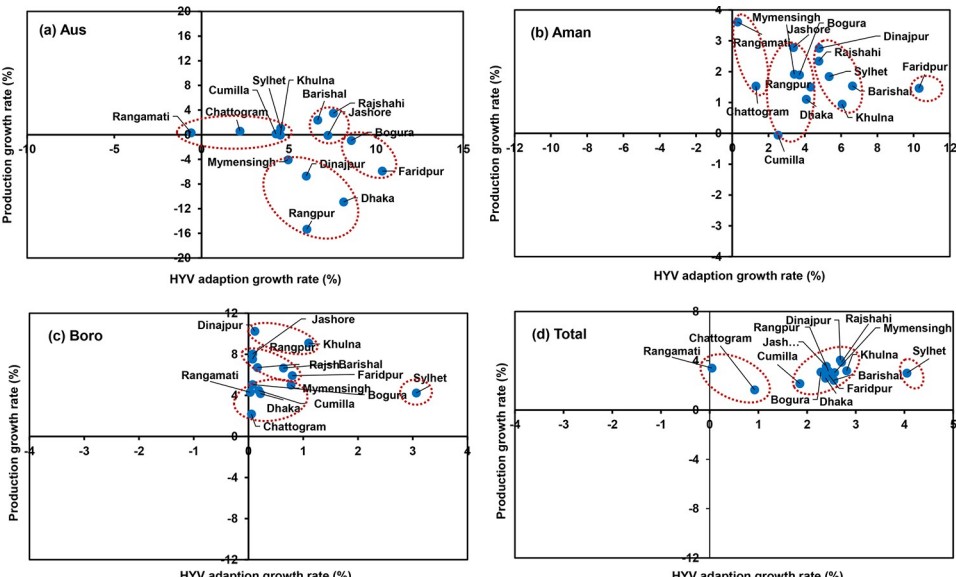

**Fig 12. Cluster analysis for grouping the regions based on high yielding variety (HYV) adoption (%) and production growth rate (%).**

Mymensingh, Bogura, Rangpur and Cumilla regions and, Rangamati and Chattogram formed a similar cluster. In the case of Boro season, Chattogram, Rangamati, Bogura, Mymensingh, Faridpur, Dhaka and Mymensingh are in the same cluster environment whereas Rajshahi, Jashore, Rangpur and Barishal are in the same cluster. The other cluster was found in Khulna and Dinajpur regions, whereas Sylhet formed a single separate cluster (Fig 12).

## Discussion

Growth and trend analysis are necessary to demonstrate the rice scenario by season, area, yield, production and varietal adaptability. Rice production depends on the season, variety, environment and geographical segmentation. Rice is cultivated in three seasons in Bangladesh namely Aus, Aman and Boro. This study revealed that the production contribution of Boro season is much higher compared to other seasons since 1998–99. Whereas production contribution of Aman season was higher from 1969–70 to 1997–98. Boro season is less vulnerable to rice disease, pests and other natural calamities and government incentives on Boro cultivation more specifically on irrigation facilities increase the Boro cultivation area [6, 18]. Moreover, after the release of two Boro mega varieties, BRRI dhan28 and BRRI dhan29 in 1994, the second silent green revolution of rice has occurred in Bangladesh. More than 60% of the area is covered by these two varieties during Boro season [19]. Currently, Boro cultivation area recorded for 61% of total cropped area in the Rabi season, which contributes 55% of total rice production in the country [18]. Our findings are also similar to this results. The maximum growing degree days, long vegetative growth, sunny weather and high amount of fertilizer utilization capacity favors the potential yield of rice during Boro season [18]. Moreover, the price of Boro rice is higher compared to other seasons which is also a key regulator of the intend of farmers to cultivate Boro rice [20]. Boro-Fallow-Fallow is the second dominant cropping pattern (13% of total net cropped area) of Bangladesh [6, 21] and this pattern profoundly found in Haor areas where other season rice cultivation is not possible except Boro due to stagnant water [22, 23]. Haor area is a major contributor (18%) of rice production in Boro season [24]

and it might be another cause of yield differentiation from other seasons. However, cold stress at seedling stage in north-west and reproductive stage in north-east (haor) regions, heat wave during flowering stage, biotic stresses mainly blast and brown plant hopper, and flash flood are becoming major challenges for Boro rice cultivation [25]. To overcome these challenges, adoption of biotic and abiotic stress tolerant varieties, precision management and irrigation infrastructure are the key production risk management strategies.

Rainfed Aman rice yield has been found as static compared to other seasons during the study period. Therefore, to ensure the future food demand of overgrowing populations, we have to sustain or increase the Aman rice production in Bangladesh. Though the rice cultivation area of Aman is higher than Aus and Boro, the yield capacity of Aman varieties is lower than the Boro varieties [5]. Climatic conditions i.e., cloudy weather, low uptake capacity of fertilizer, lowest growing degree days and short life duration are the main causes of low rice yield of Aman season. Delay planting due to anomalous rainfall, early flood, drought and other climatic hazards causes low yield of Aman rice [26, 27] which discourage farmers to cultivate rice in Aman season. Late planting of Aman sometimes hampered the rabi crops (potato, wheat, mustard, vegetables, Boro rice etc.) cultivation. Rabi crops cultivation are more profitable and safer than Aman rice and so farmers prefer to keep the land fallow during Aman instead of rice cultivation [28]. To encourage the farmers for cultivating Aman rice, short duration HYVs i.e., BRRI dhan56, BRRI dhan57, BRRI dhan71 and BRRI dhan75 could be a potential technology. Coastal area comprises 20% of the country and it covers 30% of the net cultivation area of Bangladesh [29]. To withstand the challenge of salinity of coastal region during Aman season, farmers tend to cultivate traditional local saline tolerant rice varieties which gives poor yield [30]. Prawn culture is a profitable income source to the farmers of southern part and so they prefer to culture prawn in "Gher (closed flat area)" than the rice cultivation during Aman season [31]. Deficiency of nutrients such as N, P, Cu and Zn in saline soil are also a major drawback for low yield of rice in coastal areas. However, technological advancement and development of high yielding varieties can play a vital role for increasing Aman rice production. Farmers of the southern region tend to cultivate Aman instead of Boro due to availability of soil moisture and low salinity problem. Rainfall of Aman season diminishes the salinity which favors the rice cultivation in this season [32]. Moreover, salinity is a major problem for Boro (dry season) cultivation in the southern part of Bangladesh [33]. Heavy tide is another hinder for rice cultivation in southern-coastal regions. BRRI dhan76, BRRI dhan77 and BRRI dhan78 are modern high yielding rice varieties of Bangladesh Rice Research Institute (BRRI), bred in such a way so that their seedlings will be long to withstand the tidal wetland condition [34]. These varieties give around 5 tonnes yield per hectare which is higher than the local indigenous varieties (2.5 to 3 tonnes/ha) and these varieties shed a light on increasing rice production in the coastal areas. BR23 and BRRI dhan47 are high yielding salt tolerant varieties for Aman and Boro season, respectively which covered a significant area of southern regions. Now-a-days, the newly released HYV i.e., BRRI dhan67 is gaining popularity in saline prone areas of Bangladesh.

Aus is one of the most vulnerable rice growing seasons of Bangladesh [35]. Climatic conditions i.e., hot humid weather favors the outbreak of diseases and insect pest during Aus season. Tungro is one of the severe threats of Aus production and it can cause 100% yield loss of rice under severe outbreak condition [36]. Flash flood, drought, high temperature and low yield are also a major drawback of Aus production. Moreover, late transplanting of Boro rice is one of the reasons for Aus area reduction [37]. High yield and production as well as net return of Boro rice also dampened farmers' interest to cultivate Aus rice. But intensive irrigated Boro rice cultivation depletes the underground water resulting irrigation water scarcity in the northern part of Bangladesh. So, shifting of Boro area to Aus cultivation is the key concern of the

present time. However, adaptation of modern varieties (like BRRI dhan48), disease and pest management technologies and irrigation facilities could be a good option to withstand the challenge of the critical environment of the Aus season.

Mymensingh is the largest rice producing region while the lowest rice producing region was Rangamati in terms of area and production. This is because Mymensingh is favorable for rice cultivation and Rangamati is the hilly disadvantageous areas and less access to modern technologies. We found most of the regions have positive production growth rate for Aman and Boro season. But in Aus season, especially Mymensingh, Bogura, Dinajpur, Rangpur, Dhaka and Faridpur have negative production growth rate. This is due to most of the afore-mentioned regions have dominated Boro-Fallow-T. Aman cropping pattern. In addition, those regions have intensified with non-rice cash crops and thereby the required growth dura-tion of Aus is insufficient. Noticeably, the released Aus rice varieties are not easily fit into this cropping pattern. Besides, some regions have single Boro dominated cropping patterns due to adverse agro-climatic and geographical conditions. This is why some regions have negative production growth rate. To increase the regional productivity equally, varietal and manage-ment interventions is must. Kabir et al., [5] reported that there are five unexplored areas in Bangladesh, where rice cultivation is possible. They mentioned that greater Barishal region, greater Sylhet region, South-west and greater Jashore region, Coastal *charland* in Barishal and Noakhali, and Chattogram hill tracts have unexplored areas. New HYVs, irrigation facilities, proper drainage system, and re-excavation of canals needs to be applied in these regions to increase the rice production.

Improved varieties and production technologies are the key drivers for enhancing rice pro-duction in Bangladesh. So far, we have developed 137 modern rice varieties and more than 300 production technologies by addressing different favorable and fragile ecosystems. Out of 36 stress tolerant varieties, BRRI dhan71 and BINA dhan19 in drought; BRRI dhan51, BRRI dhan52 and BINA dhan12 in submergence; BRRI dhan67, BRRI dhan97 and BINA dhan10 in saline, and BRRI dhan36 and BRRI dhan55 in cold ecosystems are the promising varieties for increasing production of stress prone areas in Bangladesh. Besides, BR11, BRRI dhan28, BRRI dhan29, BRRI dhan48, BRRI dhan49, BRRI dhan50, BRRI dhan58, BRRI dhan63, BRRI dhan81, BRRI dhan87, BRRI dhan89, BRRI dhan92, BRRI dhan96, BRRI dhan98, BRRI hybrid dhan3, BRRI hybrid dhan5 and BRRI hybrid dhan7 are the main players to boost up the rice production in Bangladesh. Use of water saving alternate wet and drying (AWD) techniques for irrigation, use of shallow water tubewell for irrigation from pond or river water, government subsidies for fertilizer and irrigation facilities, development of cropping pattern to relief the abject poverty of the northern part, disease and insect management technologies, and training of farmers to adopt modern rice cultivation techniques are acted as a catalyst to increase the rice production in Bangladesh.

## Conclusion

This study examined the area, production and yield trend, and growth rates of rice from 1984–85 to 2019–20; and analytically classified the rice production regions. Trend analyses showed an increasing and decreasing growth rates for the Aus, Aman, Boro, and total rice determi-nants in different periods. In Aus rice, area and production growth rate had significantly decreased in all the regions, while yield was significantly increased over the period. In the Aman season, the area growth rate was decreased for seven regions, but production and yield growth rates were significantly increased for all regions. Based on area, production, and yield, Boro rice have found a significant increasing trend in all the regions. In the recent period, HYVs adaption rates were found 72% for Aus, 73.5% for Aman, and 98.4% for Boro season.

During 1969–70 to 2019–20, the production contribution to the national rice production of Boro rice is significantly increasing at 0.97% per year, where Aus and Aman season production contribution significantly decreased by 0.48% and 0.49% per year, respectively. Aggregated in last 36 years rice production, Mymensingh (13.9%), Rangpur (9.8%), Bogura (8.6%), Jashore (8.6%), and Rajshahi (8.2%) were the top five production contributor regions in national rice production of Bangladesh. We have identified different cluster regions in different seasons, indicating high dissimilarities among the rice production regions. It is recommended that steps need to be taken to increase and sustain the rice production by implementing several specialized approaches. The region-wise actionable plan should be taken to rapidly adopt new technologies and highlight the research and extensions activities for fewer contributor regions to improve productivity. Cluster-wise policy strategies should be implemented for top and less contributor regions to ensure rice security as well as food security in Bangladesh.

## Supporting information

**S1 File.**
(DOCX)

**S1 Data.**
(XLSX)

## Acknowledgments

Authors express their sincere thanks to the Bangladesh Bureau of Statistics (BBS) and Bangladesh Agricultural Research Council (BARC) for making available of relevant rice data and administrative GIS shape file of Bangladesh, respectively. The authors also acknowledge several scientists of the Bangladesh Rice Research Institute for participating discussion at various stages of preparing the manuscript.

## Author Contributions

**Conceptualization:** Md. Abdullah Al Mamun, Sheikh Arafat Islam Nihad.

**Data curation:** Md. Abdullah Al Mamun, Md. Abdullah Aziz, Md. Abdul Qayum, Rokib Ahmed, Niaz Md Farhat Rahman.

**Formal analysis:** Md. Abdullah Al Mamun.

**Investigation:** Md. Abdur Rouf Sarkar, Md. Ismail Hossain, Md. Shahjahan Kabir.

**Methodology:** Md. Abdullah Al Mamun, Sheikh Arafat Islam Nihad, Md. Ismail Hossain, Md. Shahjahan Kabir.

**Resources:** Md. Abdullah Al Mamun, Md. Abdur Rouf Sarkar, Md. Abdullah Aziz, Md. Abdul Qayum, Rokib Ahmed, Niaz Md Farhat Rahman.

**Software:** Md. Abdullah Al Mamun.

**Supervision:** Md. Abdullah Al Mamun, Md. Abdur Rouf Sarkar, Md. Ismail Hossain, Md. Shahjahan Kabir.

**Validation:** Md. Abdullah Al Mamun.

**Visualization:** Md. Abdullah Al Mamun, Sheikh Arafat Islam Nihad.

**Writing – original draft:** Md. Abdullah Al Mamun, Sheikh Arafat Islam Nihad.

**Writing – review & editing:** Md. Abdullah Al Mamun, Sheikh Arafat Islam Nihad, Md. Abdur Rouf Sarkar, Md. Ismail Hossain, Md. Shahjahan Kabir.

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
