## [Decision Letter · Decision Letter 0]

11 Oct 2021

PONE-D-21-30561Growth and Trend Analysis of Area, Production and Yield of Rice: A scenario of rice security in BangladeshPLOS ONE

Dear Dr. Al Mamun,

Thank you for submitting your manuscript to PLOS ONE. After careful consideration, we feel that it has merit but does not fully meet PLOS ONE’s publication criteria as it currently stands. Therefore, we invite you to submit a revised version of the manuscript that addresses the points raised during the review process.

We look forward to receiving your revised manuscript.

Kind regards,

Vassilis G. Aschonitis

Academic Editor

PLOS ONE

Journal Requirements:

4. We note that Figures 1, 4, 6, and 8 in your submission contain [map/satellite] images which may be copyrighted. All PLOS content is published under the Creative Commons Attribution License (CC BY 4.0), which means that the manuscript, images, and Supporting Information files will be freely available online, and any third party is permitted to access, download, copy, distribute, and use these materials in any way, even commercially, with proper attribution. For these reasons, we cannot publish previously copyrighted maps or satellite images created using proprietary data, such as Google software (Google Maps, Street View, and Earth). For more information, see our copyright guidelines: http://journals.plos.org/plosone/s/licenses-and-copyright.

a. You may seek permission from the original copyright holder of Figures 1, 4, 6, and 8 to publish the content specifically under the CC BY 4.0 license.  

Reviewers' comments:

Reviewer's Responses to Questions

**Comments to the Author**

1. Is the manuscript technically sound, and do the data support the conclusions?

Reviewer #1: Yes

2. Has the statistical analysis been performed appropriately and rigorously? 

Reviewer #1: Yes

3. Have the authors made all data underlying the findings in their manuscript fully available?

Reviewer #1: Yes

4. Is the manuscript presented in an intelligible fashion and written in standard English?

Reviewer #1: Yes

5. Review Comments to the Author

Reviewer #1: The rice production of Bangadesh is the third in the world.To identify the factor underlying the trend in rice production of Bangadesh is important to carry out the region-wise plant and relative policy strategies to o ensure rice security of Bangladesh. From 1969-70 to 2019-20, rice production of Boro rice is increasing at 0.97% per year, mainly dueing to the development of two Boro mega varieties. Meanwhile, High yielding variety adoption rate of Aus and Aman has increased to 72% and 73.5%, respectively. The findings is important for rice production in Bangadesh. However, several aspects mar the overall understanding and need to be addressed:

1.The solution of Figure1, Figre4, Figure6, Figure8, Figure9 is too low to read clearly. It is necessary to provide high solution of the above Figure.

2.For Figure3(a), Figure11, Table2, Table3 and Table4, two-way-ANOVA is needed to analyze the data. A 2-way-ANOVA is a two factorial analysis (here region/season and period as the 2 parameters). With suitable post-hoc test, this should be marked with distinct statistical groups (typically different alphabets) to allow one to see if there was difference in region/season and also between periods.

3.The coefficient of variation for different regions is needed in the Table 4 to support the result “but region to region yield variations were very high” (Line 337 to 338).

4.In Figure12, for Aman and Boro，most region have positive production growth rate. But, for Mymensingh, Bogura, Dinajpur, Rangpur, Dhaka and Faridpur have negative production growth rate where the other regions have positive production growth rate. 5.More detailed discussion is needed to illustrate this phenomenon.

Minor revision: 2019-202 (Line201) should be revised as “2019-20”.

6. PLOS authors have the option to publish the peer review history of their article (what does this mean?). If published, this will include your full peer review and any attached files.

Reviewer #1: No

---

## [Author Response · Author response to Decision Letter 0]

14 Nov 2021

Editor in Chief

Dear Sir,

Thank you so much for handling our manuscript. We have tried incorporate all of your suggestions and comments into the manuscript. We also have responded the reviewers’ comments. We are also grateful for you and the reviewers to improve our manuscript. 

Response to academic editor comments

Comment 1: Please ensure that your manuscript meets PLOS ONE's style requirements, including those for file naming. 

Response 1: We have followed the manuscript requirement of PLOS ONE.

Comment 2: We suggest you thoroughly copyedit your manuscript for language usage, spelling, and grammar.

Response 2: We have edited our manuscript by the professional editor of Bangladesh Rice Research Institute. The details of the professional editor is 

Md. Abul Kashem

Technical Editor

Publication and Public Relation Division

Bangladesh Rice Research Institute

Email: head.pprd@brri.gov.bd

Edited copy of the manuscript has been uploaded as a “copyedit file” under supporting information section.

Comment 3: Data Availability statement

Response 3: The area, production and yield of rice data by seasons data from 1969-70 to 2019-20 is uploaded as a “data availability file” under supporting information section. We also revised data availability statement as per comments.

Comment 4: Copyright guidelines for figures

Response 4: GIS map describing the regional variation of area, production, yield and total rice were prepared by authors. However, administrative shape file of Bangladesh was downloaded and used from website of the Bangladesh Agricultural Research Council. Shapefile republished from Bangladesh Agricultural Research Council (BARC) database (http://maps.barcapps.gov.bd/index.php) under a CC BY license, with permission from Computer and GIS unit, BARC, original copyright 2014.

For the clarification of the copyright of maps, we added the above statements in figures 1, 4, 6 and 8 and “Data Used” section of materials and methods (Line: 127-131).

Comment 5: Review of reference list to ensure that it is complete and correct

Response 5: Reference has been corrected according to the PLOS ONE journal style. We have carefully checked in text citations.

According to the suggestions of technical editor we have added some new references in the manuscript and cited in the text accordingly.

Response to reviewers’ comments

Thank you for your insightful and valuables suggestions and comments for the improvement of our manuscript. We believe it helps to increase the scientific value of the manuscript and we are grateful for that.

Comment 5.1: The resolution of Figure 1, Figure 4, Figure 6, Figure 8, Figure 9 is too low to read clearly. It is necessary to provide high resolution of the above Figure.

Response 5.1: We have prepared and uploaded high resolution figures according to suggestions.

Comment 5.2: For Figure3(a), Figure11, Table2, Table3 and Table4, two-way-ANOVA is needed to analyze the data. A 2-way-ANOVA is a two-factorial analysis (here region/season and period as the 2 parameters). With suitable post-hoc test, this should be marked with distinct statistical groups (typically different alphabets) to allow one to see if there was difference in region/season and also between periods.

Response 5.2: 

Thank you for your valuable comments for adding the scientific value of the manuscript. Figures 3(a) and 11 were revised accordingly. But in Table 2,3 and 4 represents the exponential growth rate which contains single observation. So far our knowledge, two-way ANOVA is not suitable for this type of growth data. Therefore, for understanding the statistical variations, we have incorporated the coefficient of variation instead of two-way ANOVA in tables 2,3 and 4. 

If you have any idea for analyzing this type of data, please let us know. 

Comment 5.3: The coefficient of variation for different regions is needed in the Table 4 to support the result “but region to region yield variations were very high” (Line 337 to 338).

Response 5.3: We are grateful to for your comments. We have added coefficient of variation in the Table 4.

Comment 5.4: In Figure12, for Aman and Boro most region have positive production growth rate. But, for Mymensingh, Bogura, Dinajpur, Rangpur, Dhaka and Faridpur have negative production growth rate where the other regions have positive production growth rate. 5.More detailed discussion is needed to illustrate this phenomenon.

Response 5.4: 

We have added the following statements (Line: 460-468) according to the comments.

“We found most of the regions have positive production growth rate for Aman and Boro season. But in Aus season, especially Mymensingh, Bogura, Dinajpur, Rangpur, Dhaka and Faridpur have negative production growth rate. This is due to most of the aforementioned regions have dominated Boro-Fallow-T. Aman cropping pattern. In addition, those regions have intensified with non-rice cash crops and thereby the required growth duration of Aus is insufficient. Noticeably, the released Aus rice varieties are not easily fit into that cropping pattern. Besides, some regions have single Boro dominated cropping patterns due to adverse agro-climatic and geographical conditions. This is why some regions have negative production growth rate.”

Comment 5.5: More detailed discussion is needed to illustrate this phenomenon

Response 5.5: Discussion has been revised accordingly.

Comment 5.6: Minor revision: 2019-202 (Line201) should be revised as “2019-20”.

Response 5.6: The year is corrected accordingly.

---

## [Decision Letter · Decision Letter 1]

29 Nov 2021

Growth and trend analysis of area, production and yield of rice: A scenario of rice security in Bangladesh

PONE-D-21-30561R1

Dear Dr. Al Mamun,

We’re pleased to inform you that your manuscript has been judged scientifically suitable for publication and will be formally accepted for publication once it meets all outstanding technical requirements.

Kind regards,

Vassilis G. Aschonitis

Academic Editor

PLOS ONE

Additional Editor Comments (optional):

Reviewers' comments:

Reviewer's Responses to Questions

**Comments to the Author**

1. If the authors have adequately addressed your comments raised in a previous round of review and you feel that this manuscript is now acceptable for publication, you may indicate that here to bypass the “Comments to the Author” section, enter your conflict of interest statement in the “Confidential to Editor” section, and submit your "Accept" recommendation.

Reviewer #1: All comments have been addressed

2. Is the manuscript technically sound, and do the data support the conclusions?

Reviewer #1: Yes

3. Has the statistical analysis been performed appropriately and rigorously? 

Reviewer #1: Yes

4. Have the authors made all data underlying the findings in their manuscript fully available?

Reviewer #1: Yes

5. Is the manuscript presented in an intelligible fashion and written in standard English?

Reviewer #1: Yes

6. Review Comments to the Author

Reviewer #1: (No Response)

7. PLOS authors have the option to publish the peer review history of their article (what does this mean?). If published, this will include your full peer review and any attached files.

Reviewer #1: No

---

## [Editor Report · Acceptance letter]

2 Dec 2021

PONE-D-21-30561R1 

Growth and trend analysis of area, production and yield of rice: A scenario of rice security in Bangladesh 

Dear Dr. Al Mamun:

I'm pleased to inform you that your manuscript has been deemed suitable for publication in PLOS ONE. Congratulations! Your manuscript is now with our production department. 

Kind regards, 

on behalf of

Dr. Vassilis G. Aschonitis 

Academic Editor

PLOS ONE